# Actual versus Forecast Burden of Primary Hip and Knee Replacement Surgery in Australia: Analysis of Data from the Australian Orthopaedic Association National Joint Replacement Registry

**DOI:** 10.3390/jcm11071883

**Published:** 2022-03-28

**Authors:** Ilana N. Ackerman, Sze-Ee Soh, Richard de Steiger

**Affiliations:** 1School of Public Health and Preventive Medicine, Monash University, Melbourne, VIC 3001, Australia; sze-ee.soh@monash.edu; 2Monash-Cabrini Department of Musculoskeletal Health and Clinical Epidemiology, Melbourne, VIC 3144, Australia; 3School of Primary and Allied Health Care, Monash University, Frankston, VIC 3199, Australia; 4Epworth HealthCare, The University of Melbourne, Richmond, VIC 3121, Australia; richard.desteiger@unimelb.edu.au; 5Australian Orthopaedic Association National Joint Replacement Registry, Adelaide, SA 5000, Australia

**Keywords:** epidemiology, forecasting, hip arthroplasty, hip replacement, knee arthroplasty, knee replacement, projections

## Abstract

National projections of future joint replacement use can help us understand the changing burden of severe osteoarthritis. This study aimed to compare actual utilisation rates for primary total hip replacement (THR) and total knee replacement (TKR) to previously forecast estimates for Australia. Data from the Australian Orthopaedic Association National Joint Replacement Registry and Australian Bureau of Statistics were used to calculate ‘actual’ THR and TKR utilisation rates for the years 2014–2019, by sex and age group. ‘Forecast’ utilisation rates for 2014–2019 were derived from an earlier study that modelled two alternate scenarios for THR and TKR in Australia: Scenario 1 assumed a constant rate of surgery; Scenario 2 assumed continued growth in surgery rates. Actual utilisation rates were compared descriptively to forecast rates for females and males (overall and by age group). Rate ratios were calculated to indicate the association between actual and forecast THR and TKR rates, with a rate ratio of 1.00 reflecting perfect alignment. Over the study period, 191,996 THRs (53% in females) and 312,203 TKRs (55% in females) were performed. For both sexes, actual rates lay clearly between the Scenario 1 and 2 forecast estimates. In 2019, actual THR utilisation rates were 179 per 100,000 females (Scenario 1: 156; Scenario 2: 200) and 158 per 100,000 males (Scenario 1: 139; Scenario 2: 191). Actual TKR utilisation rates in 2019 were 289 per 100,000 females (Scenario 1: 275; Scenario 2: 387) and 249 per 100,000 males (Scenario 1: 216; Scenario 2: 312). Age-specific rate ratios were close to 1.00 for all age groups, indicating good alignment between forecast and actual joint replacement rates. These validation analyses showed that linear plus exponential growth forecasting scenarios provided an efficient approximation of actual joint replacement utilisation. This indicates our modelling techniques can be used to judiciously predict future surgery demand, including for age groups with high surgery rates.

## 1. Introduction

Primary total hip replacement (THR) and total knee replacement (TKR) are effective surgical interventions that can relieve pain and improve function for people with severe joint disease. In Australia, as in many countries, the use of this procedure has increased steadily in recent decades. The lifetime risk of primary THR in Australia is around 1 in 8 females and 1 in 10 males [1], with over 38,000 procedures performed annually [2]. Lifetime risk of primary TKR is even higher at 1 in 5 females and 1 in 7 males [3], with over 54,000 procedures performed each year [2]. Further growth in joint replacement utilisation is anticipated given an ageing population and widespread interest in maintaining physical function and quality of life into older age.

The potential future burden of joint replacement surgery can be considered by examining historical trends in surgery provision together with population size projections. This approach can help us to better understand future resource needs, including surgical workforce requirements and the likely cost to the health system, to ensure that expected demand for surgery can be met. From an epidemiology perspective, national projections of future surgery use can also help us understand the changing burden of severe hip and knee osteoarthritis. To date, projections of the anticipated future burden of THR and TKR have been published for several countries, including for the United States [4,5,6], United Kingdom [7] and Germany [8]. We have previously forecast the future burden of primary THR and TKR for osteoarthritis in Australia to the year 2030 using two different modelling scenarios [9]. The availability of actual joint replacement utilisation data for the initial six years of the forecast period allows us to now assess the accuracy of these projections. This study aimed to compare actual rates of primary THR and primary TKR in 2014–2019 with the forecast rates for these years.

## 2. Materials and Methods

### 2.1. Study Design

Secondary data analysis of available national-level joint replacement and population data.

### 2.2. Data Sources

We obtained non-identifiable, aggregate data on all primary THR procedures performed for osteoarthritis and all primary TKR procedures performed for osteoarthritis in Australia from 2014 to 2019 from the Australian Orthopaedic Association National Joint Replacement Registry (AOANJRR). These data were stratified by year of surgery, age group, sex and hospital setting (public or private). The AOANJRR collects data on all joint replacement procedures performed in Australia, with very high levels of data completeness that remained stable over the study period (>98% capture of joint replacements in 2014, >98% in 2015, >98% in 2016, >99% in 2017, >98% in 2018 and >98% in 2019). The AOANJRR data are validated against health department records using a sequential multi-level matching process to ensure accuracy [2]. These percentages reflect the number of joint replacements captured by the AOANJRR that can be matched to health department records (via initial reporting and verification to identify and retrieve any unmatched procedures, based on ICD-10 codes), expressed as a proportion of all Australian joint replacement procedures received by the AOANJRR.

Data on Australia’s population structure by age and sex for the years 2014–2019 were obtained from the Australian Bureau of Statistics [10]. Population data were limited to people aged 15 years and over, given relatively few joint replacements are performed for people below this age. In Australia, joint replacement surgery is performed in public and private hospital settings. Surgery at public hospitals is covered under the taxpayer-funded Medicare scheme, while surgery at private hospitals is largely reimbursed by private health insurance. The cost of primary THR and TKR procedures in the public hospital system was extracted from the most recent National Hospital Cost Data Collection (Round 23, financial years 2018–2019) [11]. Codes I33A and I33B were averaged to derive a cost per THR of AUD23,375, and codes I04A and I04B were averaged to derive a cost per TKR of AUD22,043. The cost of primary THR and TKR procedures in the private hospital system was extracted from the consumer websites of two major Australian private health insurers [12,13]. These estimates were averaged to derive a cost of AUD26,523 per THR and AUD25,276 per TKR.

The forecast number and rate of primary THR and primary TKR for the years 2014–2019 was previously modelled according to two potential scenarios [9]:Age-specific and sex-specific primary THR and primary TKR rates in 2013 were projected to continue at a constant rate until the year 2030 (‘Scenario 1’). Scenario 1 was therefore driven purely by expected population growth and ageing.Age-specific and sex-specific primary THR and primary TKR rates were projected to continue to increase as they had from 2003 to 2013, using Poisson regression analysis with age group, sex and procedure year included as model covariates (‘Scenario 2’). Scenario 2 was therefore driven by an expected increase in surgery rates as well as expected population growth and ageing.

### 2.3. Data Analysis

All analyses were performed using Microsoft Excel 2019 (version 1808). Actual utilisation rates for primary THR and primary TKR were calculated for 2014–2019 by dividing the number of actual procedures in each year by the relevant Australian population for that year. These are reported as actual utilisation rates per 100,000 population, with separate calculations undertaken for males and females, and by age group. Where bilateral THR or TKR procedures were performed in the same year (either simultaneously or sequentially), these were counted as two procedures to avoid underestimating utilisation rates. Actual utilisation rates for primary THR and TKR were compared descriptively to the forecast utilisation rates for 2014–2019. Rate ratios (the ratio between the actual utilisation rate and the forecast rate for each age- and sex-specific group) were also calculated to indicate the strength of the association between the two rates. The cost of primary THR and primary TKR for the public health system, private health system and overall (public plus private health systems) over the study period was estimated, based on the number of procedures each year and the proportion of procedures that were performed in public or private settings. All costs are reported in Australian dollars (1 AUD is approximately 0.74 USD).

## 3. Results

### 3.1. Characteristics of the Cohort

From 2014 to 2019, 191,996 primary THR procedures and 312,203 primary TKR procedures were performed in Australia. Table 1 summarises the demographic characteristics of patients who received THR or TKR during the study period. Reflecting the nature of joint replacement provision in Australia, most THR and TKR procedures (70%) were performed in the private hospital sector. The estimated overall cost to the health system for 2014–2019 was AUD4.91 billion for THR and AUD7.59 billion for TKR (Table 1).

### 3.2. Actual versus Forecast Rates of Total Hip Replacement

Growth in the rate of THR for females progressed according to forecast Scenario 2 from 2014 to 2016, after which point the rate of THR plateaued but remained within the two forecast scenarios (Figure 1). In 2019, the rate of THR for females was 179 per 100,000 population, compared to 156 per 100,000 for Scenario 1 and 200 per 100,000 population for Scenario 2. For males, the rate of THR also closely followed the Scenario 2 forecast until 2016 (Figure 2). After that time, the THR rate remained relatively steady to reach 158 per 100,000 population in 2019 (compared to 139 per 100,000 population for Scenario 1 and 191 per 100,000 population for Scenario 2).

### 3.3. Actual versus Forecast Rates of Total Knee Replacement

The rate of TKR for females was most closely aligned with Scenario 1 from 2014 to 2019, with little growth evident over this period (Figure 3). In 2019, the actual TKR rate was 289 per 100,000 females, compared to 275 per 100,000 (Scenario 1) and 387 per 100,000 (Scenario 2). For males, the rate of TKR initially increased according to Scenario 2, after which time less growth was seen (Figure 4). The rate of TKR for males in 2019 was 249 per 100,000 population, compared to 216 per 100,000 population (Scenario 2) and 312 per 100,000 population (Scenario 2).

### 3.4. Age-Specific Analyses

Table 2 presents the rate of THR procedures for females by age group. For the under 40 age group, the actual THR rate aligned closely with the Scenario 2 forecast until 2018, before decreasing in 2019. For the 40–69 age group, the actual rate of THR was aligned with the Scenario 2 projections until 2017 when the THR rate decreased slightly before increasing again in 2018. Growth in THR rates for the 70 years and over age group was comparable to the Scenario 2 forecast for 2014–2017, after which time the rate of THR decreased before rising again in 2019.

Table 3 shows the forecast versus actual THR rates for males. For the under 40 age group, the actual rate of THR slightly exceeded the Scenario 2 forecast for each year. For the 40–69 age group, actual THR rates initially exceeded the Scenario 2 estimates for 2014–2016 before falling within the Scenario 1 and Scenario 2 estimates for 2017–2019. For the oldest age group, actual THR rates lay between the Scenario 1 and Scenario 2 forecast estimates for each year.

Table 4 presents the age-specific rate of TKR procedures for females. Procedure rates for the under 40 age group remained relatively close to the Scenario 1 estimates each year. For the 40–69 age group, TKR rates tracked according to the Scenario 2 forecast until 2016 before plateauing. For the 70 years and over group, actual TKR rates remained above the Scenario 1 estimates, to a small degree, at all time points.

As shown in Table 5, the actual rate of TKR procedures for males aged under 40 years was similar to the Scenario 1 forecast estimates. For males aged 40–69 years, TKR rates were closest to the Scenario 2 estimates until 2017, when the number of procedures plateaued. For those aged 70 years and over, actual TKR rates were most closely aligned with the Scenario 1 estimates.

## 4. Discussion

This study found that actual THR and TKR rates for both sexes fell clearly between two forecasting scenarios, with no obvious outliers. Overall, the number of THR procedures for females and males grew by 23% and 20%, respectively, from 2014 to 2019, while the number of TKR procedures increased by 17% for females and 24% for males over this period. This compares to forecast growth of 15% (based on Scenario 1) to 39% (based on Scenario 2) for THR and 15% (Scenario 1) to 48% (Scenario 2) for TKR. This shows that our forecasting approach (which considered a constant rate of surgery scenario plus a continued surgery growth scenario, in combination with national estimates of population growth and ageing) was able to appropriately quantify potential uncertainty in joint replacement rates.

We uniquely compared forecast estimates for joint replacement surgery in Australia to actual surgery utilisation, to examine the accuracy of these projections and inform future modelling techniques. The ability to generate a realistic estimation of the future use of joint replacement surgery is important for several reasons. Firstly, such projections can support healthcare resource planning in the context of ageing populations that are more likely to require joint replacement. Secondly, an understanding of expected surgery growth can inform new models of pre- and post-operative care to ensure that health systems can efficiently meet anticipated demand. Thirdly, joint replacement numbers provide a population-level indicator of osteoarthritis disease burden—unsustainable growth can signal a need for greater investment in prevention and early management strategies. However, it is important to recognise that surgery projections are based on available data and a set of assumptions. As such, there is always inherent uncertainty in the forecast estimates. This highlights the importance of (a) using multiple modelling scenarios, and (b) comparing forecast estimates with ‘real data’, once these become available, to examine model accuracy. We are only aware of two other studies, both from the United States, that have sought to revisit joint replacement projections. Both studies used a sample of hospital discharge records (the National Inpatient Sample) rather than complete population-level arthroplasty data as the basis for their forecast estimates. In the context of national economic challenges, Kurtz et al. [4] published updated forecast estimates for THR and TKR which incorporated health expenditure metrics. They found reasonable agreement between their initial projections [14] and national administrative data, as well as between the updated projections and national administrative data [4]. More recently, Sloan et al. examined the accuracy of linear models and exponential growth projection models with respect to future joint replacement surgery [5]. The linear models (akin to our Scenario 1 surgery projections) were found to produce the best approximation of actual THR and TKR use over a 15-year period. Our data also show that while overall joint replacement rates in Australia did increase for both sexes over a six-year period, the rates did not increase exponentially. In particular, TKR rates for females tracked only slightly above our conservative projection scenario. The potential reasons for this are unclear as rates of obesity (a major driver of knee osteoarthritis) for Australian females have not reduced over time [15]. Rather, this finding might reflect achievement of a balance between unmet need and receipt of TKR for females. Our age-specific analyses showed that rate ratios were closest to 1.00 for both the 40–69 and 70 years and over age groups, indicating the best alignment of projected and actual surgery rates. The higher and lower rate ratios for the under 40 age group likely reflects the very low rates of joint replacement surgery for younger patients in Australia (at less than 5 THR procedures per 100,000 population and less than 1 TKR procedure per 100,000 population), and we note that absolute differences in forecast rates versus actual rates for this age group were only small.

Rates of joint replacement are usually based on a numerator (the number of procedures) and a denominator (the relevant population size). The number of procedures is commonly sourced from administrative datasets or national arthroplasty registries and the relevant population size is frequently sourced from national statistical agencies. For our published forecast estimates [9], we used available population size projections from the Australian Bureau of Statistics for the years 2014 onwards. These population projections are based on a series of assumptions around migration, fertility and life expectancy. When calculating the actual rates of surgery for the present study, we used actual population size data from the Australian Bureau of Statistics that are derived from the national census. Comparing the population size data from these two sources for the years 2014–2019 shows that while the population size projections for females were highly accurate, the population size projections for males aged 40–69 years overestimated the actual population size for this group (Appendix A). This overestimation would have impacted the surgical rate estimates for this age group (as well as for the overall estimates for males) to a small degree. For example, using the projected population size as the denominator produced an overall rate of THR for males in 2019 of 156 per 100,000 population versus 158 per 100,000 using the actual population size as the denominator. Similarly, the overall rate of TKR for males in 2019 would have been 247 per 100,000 using the projected population size rather than 249 per 100,000 when using the actual population size. For completeness, we have also reported absolute numbers of joint replacement, although these do not allow for population growth over time to be considered.

This study had several key strengths, including the use of joint replacement data from a longstanding national registry with excellent population coverage. We included THR and TKR procedures performed in public and private hospital settings to reflect usual practice in Australia. Joint replacement rates were examined for younger as well as older age groups, and sex-specific analyses were also undertaken, given some differences in surgery rates between males and females. We also acknowledge the study limitations. There may have been local policy changes that impacted joint replacement provision during the study period of which we are unaware. We did not analyse data for the year 2020 due to the major impacts of the COVID-19 pandemic on elective surgery provision, with 7086 fewer joint replacements performed in Australia in 2020 compared to the previous year [2]. It is highly likely that previous forecasts for the year 2020 onwards will not align with actual joint replacement provision given ongoing COVID-related surgery restrictions in some Australian states and the significant ‘catch up’ that will be required in future years. Once non-urgent surgery is permitted to resume without restrictions, it is possible that surgery rates will grow rapidly, as initially forecast in our Scenario 2 modelling. 

## 5. Conclusions

This study has shown that two alternative forecasting scenarios (one conservative and one exponential) provided an efficient approximation of actual joint replacement utilisation. The modelling produced competent estimates for both sexes and for age groups with low or high surgery rates. These analyses support the use of our modelling techniques based on historical trends in joint replacement rates and population projections to predict future surgery demand and inform future resource and service planning requirements.

## Figures and Tables

**Figure 1 jcm-11-01883-f001:**
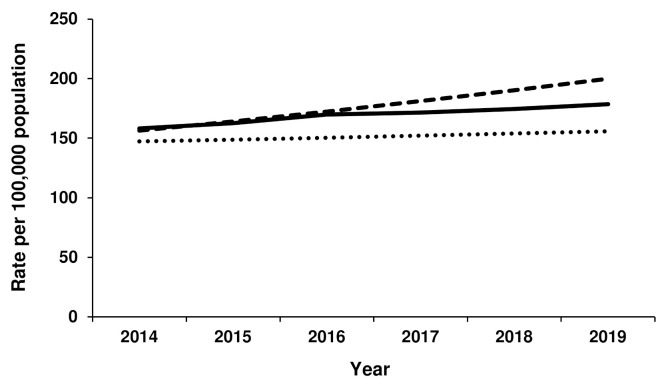
Actual versus forecast total hip replacement rates for females. Dotted line represents forecast Scenario 1, dashed line represents forecast Scenario 2 and solid line represents actual rates for 2014–2019.

**Figure 2 jcm-11-01883-f002:**
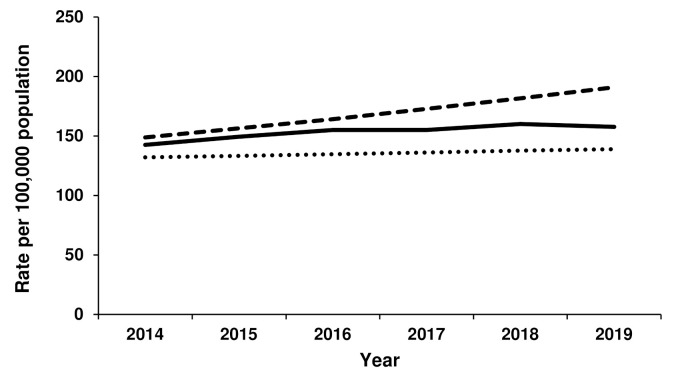
Actual versus forecast total hip replacement rates for males. Dotted line represents forecast Scenario 1, dashed line represents forecast Scenario 2 and solid line represents actual rates for 2014–2019.

**Figure 3 jcm-11-01883-f003:**
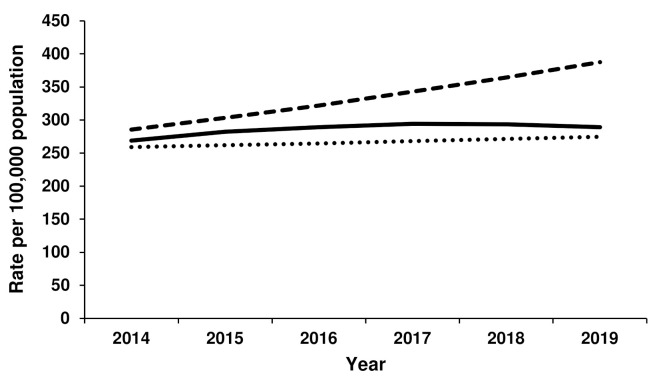
Actual versus forecast total knee replacement rates for females. Dotted line represents forecast Scenario 1, dashed line represents forecast Scenario 2 and solid line represents actual rates for 2014–2019.

**Figure 4 jcm-11-01883-f004:**
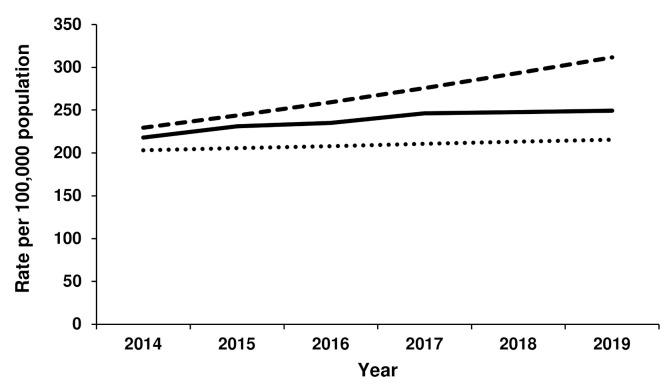
Actual versus forecast total knee replacement rates for males. Dotted line represents forecast Scenario 1, dashed line represents forecast Scenario 2 and solid line represents actual rates for 2014–2019.

**Table 1 jcm-11-01883-t001:** Characteristics of joint replacement surgery in Australia for 2014–2019.

Characteristic	Total Hip Replacement (*n* = 191,996)	Total Knee Replacement (*n* = 312,203)
Females, *n* (%)	102,241 (53)	172,846 (55)
Age group, *n* (%)		
<40 years	1851 (1)	299 (<1)
40–69 years	100,898 (53)	171,319 (55)
≥70 years	89,247 (46)	140,585 (45)
Hospital sector, *n* (%)		
Private	134,302 (70)	218,485 (70)
Public	57,694 (30)	93,718 (30)
Estimated total cost		
Private	AUD3.56 billion	AUD5.52 billion
Public	AUD1.35 billion	AUD2.07 billion

AUD: Australian dollars.

**Table 2 jcm-11-01883-t002:** Forecast versus actual rates of total hip replacement for females, by age group.

Age Group	Year	Forecast Scenario 1		Forecast Scenario 2		Actual		Rate Ratio *	
		Number	Rate	Number	Rate	Number	Rate	Scenario 1	Scenario 2
<40 years	2014	97	2.4 per 100,000	114	2.8 per 100,000	131	3.2 per 100,000	1.34	1.15
	2015	99	2.4 per 100,000	121	3.0 per 100,000	120	2.9 per 100,000	1.21	0.99
	2016	102	2.4 per 100,000	129	3.1 per 100,000	110	2.6 per 100,000	1.08	0.85
	2017	104	2.5 per 100,000	137	3.2 per 100,000	134	3.2 per 100,000	1.28	0.97
	2018	107	2.5 per 100,000	146	3.4 per 100,000	158	3.7 per 100,000	1.47	1.07
	2019	109	2.5 per 100,000	155	3.6 per 100,000	126	2.9 per 100,000	1.15	0.80
40–69 years	2014	7129	164.3 per 100,000	7648	176.2 per 100,000	7656	176.6 per 100,000	1.07	1.00
	2015	7296	165.7 per 100,000	8130	184.6 per 100,000	7984	181.8 per 100,000	1.10	0.98
	2016	7464	167.2 per 100,000	8639	193.5 per 100,000	8495	191.0 per 100,000	1.14	0.99
	2017	7554	167.5 per 100,000	9080	201.3 per 100,000	8245	183.8 per 100,000	1.10	0.91
	2018	7680	168.4 per 100,000	9588	210.2 per 100,000	8566	189.2 per 100,000	1.12	0.90
	2019	7820	169.3 per 100,000	10,139	219.5 per 100,000	8725	190.6 per 100,000	1.13	0.87
70+ years	2014	6984	544.4 per 100,000	7334	571.7 per 100,000	7474	584.5 per 100,000	1.07	1.02
	2015	7207	544.7 per 100,000	7862	594.2 per 100,000	7824	595.5 per 100,000	1.09	1.00
	2016	7444	545.3 per 100,000	8434	617.9 per 100,000	8335	616.3 per 100,000	1.13	1.00
	2017	7796	547.4 per 100,000	9173	644.1 per 100,000	9018	639.4 per 100,000	1.17	0.99
	2018	8111	549.1 per 100,000	9911	670.9 per 100,000	9269	633.2 per 100,000	1.15	0.94
	2019	8418	550.7 per 100,000	10,680	698.7 per 100,000	9871	650.4 per 100,000	1.18	0.93

* Ratio of rate (actual) to rate (forecast); rate of 1.00 indicates perfect alignment.

**Table 3 jcm-11-01883-t003:** Forecast versus actual rates of total hip replacement for males, by age group.

Age Group	Year	Forecast Scenario 1		Forecast Scenario 2		Actual		Rate Ratio *	
		Number	Rate	Number	Rate	Number	Rate	Scenario 1	Scenario 2
<40 years	2014	128	3.1 per 100,000	116	2.8 per 100,000	149	3.6 per 100,000	1.17	1.28
	2015	131	3.1 per 100,000	124	2.9 per 100,000	159	3.8 per 100,000	1.22	1.29
	2016	134	3.2 per 100,000	132	3.1 per 100,000	199	4.7 per 100,000	1.49	1.53
	2017	138	3.2 per 100,000	140	3.2 per 100,000	176	4.1 per 100,000	1.28	1.26
	2018	142	3.2 per 100,000	150	3.4 per 100,000	199	4.5 per 100,000	1.41	1.33
	2019	145	3.3 per 100,000	159	3.6 per 100,000	190	4.3 per 100,000	1.31	1.19
40–69 years	2014	7258	170.7 per 100,000	7581	178.3 per 100,000	7849	185.7 per 100,000	1.09	1.04
	2015	7398	171.6 per 100,000	8035	186.4 per 100,000	8293	194.3 per 100,000	1.13	1.04
	2016	7535	172.6 per 100,000	8510	194.9 per 100,000	8709	202.2 per 100,000	1.17	1.02
	2017	7604	172.7 per 100,000	8918	202.5 per 100,000	8649	199.6 per 100,000	1.16	0.99
	2018	7704	173.1 per 100,000	9387	210.9 per 100,000	8923	204.5 per 100,000	1.18	0.97
	2019	7822	173.6 per 100,000	9904	219.9 per 100,000	8804	200.0 per 100,000	1.15	0.91
70+ years	2014	5061	482.4 per 100,000	6336	604.0 per 100,000	5382	514.9 per 100,000	1.07	0.85
	2015	5259	482.1 per 100,000	6839	627.0 per 100,000	5762	531.5 per 100,000	1.10	0.85
	2016	5471	482.1 per 100,000	7389	651.2 per 100,000	6071	538.8 per 100,000	1.12	0.83
	2017	5771	483.1 per 100,000	8093	677.5 per 100,000	6395	538.7 per 100,000	1.12	0.80
	2018	6037	483.7 per 100,000	8792	704.4 per 100,000	6845	551.0 per 100,000	1.14	0.78
	2019	6285	484.1 per 100,000	9506	732.3 per 100,000	7001	540.6 per 100,000	1.12	0.74

* Ratio of rate (actual) to rate (forecast); rate of 1.00 indicates perfect alignment.

**Table 4 jcm-11-01883-t004:** Forecast versus actual rates of total knee replacement for females, by age group.

Age Group	Year	Predicted Scenario 1		Predicted Scenario 2		Actual		Rate Ratio *	
		Number	Rate	Number	Rate	Number	Rate	Scenario 1	Scenario 2
<40 years	2014	25	0.6 per 100,000	36	0.9 per 100,000	21	0.5 per 100,000	0.83	0.58
	2015	26	0.6 per 100,000	39	0.9 per 100,000	26	0.6 per 100,000	1.00	0.67
	2016	27	0.6 per 100,000	42	1.0 per 100,000	29	0.7 per 100,000	1.09	0.70
	2017	27	0.6 per 100,000	45	1.1 per 100,000	29	0.7 per 100,000	1.05	0.65
	2018	28	0.7 per 100,000	48	1.1 per 100,000	21	0.5 per 100,000	0.74	0.43
	2019	29	0.7 per 100,000	52	1.2 per 100,000	25	0.6 per 100,000	0.85	0.48
40–69 years	2014	13,472	310.4 per 100,000	14,061	324.0 per 100,000	14,056	324.2 per 100,000	1.04	1.00
	2015	13,797	313.3 per 100,000	15,131	343.6 per 100,000	15,012	341.8 per 100,000	1.09	0.99
	2016	14,125	313.2 per 100,000	16,276	360.9 per 100,000	15,763	354.4 per 100,000	1.13	0.98
	2017	14,304	317.2 per 100,000	17,303	383.7 per 100,000	16,106	359.0 per 100,000	1.13	0.94
	2018	14,553	319.0 per 100,000	18,489	405.3 per 100,000	15,825	349.6 per 100,000	1.10	0.86
	2019	14,828	321.1 per 100,000	19,786	428.4 per 100,000	15,634	341.5 per 100,000	1.06	0.80
70+ years	2014	11,502	896.5 per 100,000	13,457	1,048.9 per 100,000	11,867	928.0 per 100,000	1.04	0.88
	2015	11,887	898.5 per 100,000	14,596	1,103.2 per 100,000	12,640	962.0 per 100,000	1.07	0.87
	2016	12,295	900.7 per 100,000	15,844	1,160.7 per 100,000	13,016	962.4 per 100,000	1.07	0.83
	2017	12,911	906.6 per 100,000	17,452	1,225.5 per 100,000	13,722	973.0 per 100,000	1.07	0.79
	2018	13,457	911.0 per 100,000	19,089	1,292.2 per 100,000	14,414	984.8 per 100,000	1.08	0.76
	2019	13,980	914.5 per 100,000	20,818	1,361.8 per 100,000	14,640	964.7 per 100,000	1.05	0.71

* Ratio of rate (actual) to rate (forecast); rate of 1.00 indicates perfect alignment.

**Table 5 jcm-11-01883-t005:** Forecast versus actual rates of total knee replacement for males, by age group.

Age Group	Year	Predicted Scenario 1		Predicted Scenario 2		Actual		Rate Ratio *	
		Number	Rate	Number	Rate	Number	Rate	Scenario 1	Scenario 2
<40 years	2014	25	0.6 per 100,000	31	0.7 per 100,000	20	0.5 per 100,000	0.79	0.65
	2015	26	0.6 per 100,000	33	0.8 per 100,000	20	0.5 per 100,000	0.78	0.61
	2016	27	0.6 per 100,000	36	0.8 per 100,000	27	0.6 per 100,000	1.02	0.77
	2017	27	0.6 per 100,000	38	0.9 per 100,000	27	0.6 per 100,000	0.99	0.71
	2018	28	0.6 per 100,000	41	0.9 per 100,000	27	0.6 per 100,000	0.95	0.65
	2019	29	0.7 per 100,000	45	1.0 per 100,000	27	0.6 per 100,000	0.92	0.60
40-69 years	2014	10,859	255.3 per 100,000	11,690	274.9 per 100,000	11,641	275.4 per 100,000	1.08	1.00
	2015	11,091	257.3 per 100,000	12,539	290.9 per 100,000	12,670	296.8 per 100,000	1.15	1.02
	2016	11,320	259.3 per 100,000	13,439	307.9 per 100,000	13,088	303.9 per 100,000	1.17	0.99
	2017	11,427	259.4 per 100,000	14,241	323.3 per 100,000	13,689	315.8 per 100,000	1.22	0.98
	2018	11,590	260.4 per 100,000	15,165	340.7 per 100,000	13,924	319.1 per 100,000	1.23	0.94
	2019	11,782	261.5 per 100,000	16,188	359.3 per 100,000	13,911	316.0 per 100,000	1.21	0.88
70+ years	2014	8274	788.7 per 100,000	9902	943.8 per 100,000	8798	841.7 per 100,000	1.07	0.89
	2015	8602	788.7 per 100,000	10,809	991.1 per 100,000	9310	858.8 per 100,000	1.09	0.87
	2016	8952	788.9 per 100,000	11,811	1,040.9 per 100,000	9581	850.3 per 100,000	1.08	0.82
	2017	9449	791.0 per 100,000	13,091	1,095.8 per 100,000	10,460	881.1 per 100,000	1.11	0.80
	2018	9889	792.3 per 100,000	14,385	1,152.6 per 100,000	10,768	866.8 per 100,000	1.09	0.75
	2019	10,295	793.1 per 100,000	15,730	1,211.7 per 100,000	11,369	877.9 per 100,000	1.11	0.72

* Ratio of rate (actual) to rate (forecast); rate of 1.00 indicates perfect alignment.

## Data Availability

Actual joint replacement data are not publicly available under the current ethics approval. A summary of the joint replacement projections can be obtained from: Ackerman I.N., Bohensky M.A., Zomer E., et al. The projected burden of primary total knee and hip replacements for osteoarthritis in Australia through the year 2030. *BMC Musculoskeletal Disorders* **2019**, *20*, 90 [9].

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
