# Peer review of "Actual versus Forecast Burden of Primary Hip and Knee Replacement Surgery in Australia: Analysis of Data from the Australian Orthopaedic Association National Joint Replacement Registry"

_jcm, 2022, doi:10.3390/jcm11071883_

Round 1
Reviewer 1 Report
The authors compared estimated to actual utilisation rates of THR and TKR in Australia. Their findings support the notion that linear and exponential growth forecasting scenarios provide good approximations of actual joint replacement utilisation.
Title: Good
Abstract: Good
Introduction: Good
Methods: Good
line 73-4: choose one: "via" OR "using"
line 86-88: Combine the two sentences: "The forecast number and rate of primary THR and primary TKR for the years 2014-2019 was previously modelled according to two potential scenarios:"
Line 100: "analyses were performed using Microsoft Excel (ver:??)"
Line 101: replace "count" with "number"
Results: OK
Line 117-26: Shorten. Rely on the Table to convey the information and do not repeat the same data in the text.
Line 165-91: Shorten. Rely on the Table to convey the information and do not repeat the same data in the text.
Discussion: OK
Line 201-3: Do not restate the purpose or a version thereof, but rather start with the main finding of the study.
Line 205: What is an "appropriate 'band of uncertainty'"
Line 201-11. The clinical relevance/impact of the main finding should be clearly stated rather than rehashing the results.
Conclusion: OK
Reviewer 2 Report
Thank you for the opportunity to review this paper. The topic is highly relevant and up to date.
The title and abstract cover the main aspect of the work. In addition, the introduction provide background relevant to the study.
The results are presented in a readable format with additional clarification provided through tables and figures. This study is valuable to the reader as it clearly demonstrate the increasing burden are not exponentially!
Methods
a) The authors do not state completeness of primary HA and KA . Please provide percentages for this statement, and explanation on how these numbers are calculated. In addition, was completeness stable over the study period.
b) Please provide an argument for applying < 15 years as exclusion criteria and not 40 years. Or when do we expect osteoarthritis being the prominent cause of joint replecement. How many patients were excluded due to this?
Results
a) Most replacements are performed at private hospitals, 70%. There is a difference i costs bwteeen public and private hospitals. Who reimburse these costs, the patients or national health? This may influence the increasing incidence depending on who pays or the local/national economy. It may explain the 2016 point of change seen in figure 1 and changes in figure 2, 3 and 4?
Discussion
Well written and provide excellent summary of this problem of how to estimate the future burden.
Reviewer 3 Report
This study described two alternative forecasting scenarios for diffusion of THA and TKA in the next future. The bias of the COVID pandemics has been taken into account. The source of the data is sound and reliable.
As the authors mentioned, these analyses support the use of modelling techniques based on historical trends in joint replacement rates and population projections to predict future surgery demand and inform future resource and service planning requirements.
